# Assessment of Nutritional Status and Nutrition Impact Symptoms in Patients Undergoing Resection for Upper Gastrointestinal Cancer: Results from the Multi-Centre NOURISH Point Prevalence Study

**DOI:** 10.3390/nu13103349

**Published:** 2021-09-24

**Authors:** Irene Deftereos, Justin M. C. Yeung, Janan Arslan, Vanessa M. Carter, Elizabeth Isenring, Nicole Kiss

**Affiliations:** 1Department of Surgery, Western Precinct, Melbourne Medical School, The University of Melbourne, St Albans, VIC 3021, Australia; justin.yeung@unimelb.edu.au (J.M.C.Y.); janan.arslan@unimelb.edu.au (J.A.); 2Department of Nutrition and Dietetics, Western Health, Footscray, VIC 3011, Australia; Vanessa.carter@wh.org.au; 3Department of Colorectal Surgery, Western Health, Footscray, VIC 3011, Australia; 4Western Health Chronic Disease Alliance, Western Health, Footscray, VIC 3011, Australia; 5Faculty of Health Sciences and Medicine, Bond University, Robina, QLD 4226, Australia; Lisenrin@bond.edu.au; 6Department of Nutrition and Dietetics, Princess Alexandra Hospital, Brisbane, QLD 4102, Australia; 7Institute for Physical Activity and Nutrition, Deakin University, Geelong, VIC 3220, Australia; nicole.kiss@deakin.edu.au; 8Allied Health Research, Peter MacCallum Cancer Centre, Melbourne, VIC 3000, Australia

**Keywords:** gastrointestinal cancer, gastrointestinal surgery, subjective global assessment, malnutrition, nutrition impact symptoms

## Abstract

Background: Identification and treatment of malnutrition are essential in upper gastrointestinal (UGI) cancer. However, there is limited understanding of the nutritional status of UGI cancer patients at the time of curative surgery. This prospective point prevalence study involving 27 Australian tertiary hospitals investigated nutritional status at the time of curative UGI cancer resection, as well as presence of preoperative nutrition impact symptoms, and associations with length of stay (LOS) and surgical complications. Methods: Subjective global assessment, hand grip strength (HGS) and weight were performed within 7 days of admission. Data on preoperative weight changes, nutrition impact symptoms, and dietary intake were collected using a purpose-built data collection tool. Surgical LOS and complications were also recorded. Multivariate regression models were developed for nutritional status, unintentional weight loss, LOS and complications. Results: This study included 200 patients undergoing oesophageal, gastric and pancreatic surgery. Malnutrition prevalence was 42% (95% confidence interval (CI) 35%, 49%), 49% lost ≥5% weight in 6 months, and 47% of those who completed HGS assessment had low muscle strength with no differences between surgical procedures (*p* = 0.864, *p* = 0.943, *p* = 0.075, respectively). The overall prevalence of reporting at least one preoperative nutrition impact symptom was 55%, with poor appetite (37%) and early satiety (23%) the most frequently reported. Age (odds ratio (OR) 4.1, 95% CI 1.5, 11.5, *p* = 0.008), unintentional weight loss of ≥5% in 6 months (OR 28.7, 95% CI 10.5, 78.6, *p* < 0.001), vomiting (OR 17.1, 95% CI 1.4, 207.8, 0.025), reduced food intake lasting 2–4 weeks (OR 7.4, 95% CI 1.3, 43.5, *p* = 0.026) and ≥1 month (OR 7.7, 95% CI 2.7, 22.0, *p* < 0.001) were independently associated with preoperative malnutrition. Factors independently associated with unintentional weight loss were poor appetite (OR 3.7, 95% CI 1.6, 8.4, *p* = 0.002) and degree of solid food reduction of <75% (OR 3.3, 95% CI 1.2, 9.2, *p* = 0.02) and <50% (OR 4.9, 95% CI 1.5, 15.6, *p* = 0.008) of usual intake. Malnutrition (regression coefficient 3.6, 95% CI 0.1, 7.2, *p* = 0.048) and unintentional weight loss (regression coefficient 4.1, 95% CI 0.5, 7.6, *p* = 0.026) were independently associated with LOS, but no associations were found for complications. Conclusions: Despite increasing recognition of the importance of preoperative nutritional intervention, a high proportion of patients present with malnutrition or clinically significant weight loss, which are associated with increased LOS. Factors associated with malnutrition and weight loss should be incorporated into routine preoperative screening. Further investigation is required of current practice for dietetics interventions received prior to UGI surgery and if this mitigates the impact on clinical outcomes.

## 1. Introduction

Upper gastrointestinal (UGI) cancers carry some of the highest mortality rates of all cancers worldwide [1]. Although there have been advances in treatments and associated survival rates, surgery for UGI cancer carries a high morbidity rate and is considered a major procedure [2,3,4]. Optimisation of nutritional status is recognised as a key component of perioperative care in major abdominal surgery, as malnutrition and unintentional weight loss can increase the risk of surgical complications, length of stay (LOS) and mortality [5,6].

Patients with UGI cancer are one of the highest-risk groups for malnutrition [7]. Although oncological care pathways advocate for the dietitian to be an essential member of the multi-disciplinary treatment team [8], often there is a lack of resourcing to support adequate nutritional screening and intervention prior to gastrointestinal cancer surgery [9,10]. Furthermore, there is a lack of high-quality evidence regarding optimal methods of nutrition support prior to UGI cancer surgery and in neoadjuvant therapy [11], and UGI cancer-specific evidence-based nutrition guidelines do not exist [12]. Previous studies have investigated the nutritional status of UGI cancer patients in the ambulatory and inpatient settings, with reported malnutrition prevalence between 48 and 52% depending on the assessment method utilised [13,14,15]. More recently, the INFORM study has investigated the nutritional status of oesophageal and head and neck cancer populations [16]. However, there is a lack of data regarding the nutritional status of UGI cancer patients at the time of curative surgery. Furthermore, little is known regarding preoperative factors that are associated with malnutrition or clinically significant weight loss on presentation for surgery.

The Nutritional Outcomes of patients Undergoing Resection for upper gastroIntestinal cancer in AuStralian Hospitals study (The NOURISH Point Prevalence Study) was conducted to investigate nutritional status and nutritional interventions received by patients undergoing UGI cancer surgery and associations with clinical outcomes, as well as health-service-level practices [17]. NOURISH is the largest study internationally to assess nutritional status in UGI cancer patients at the time of surgery using a validated assessment method [17]. The aims of this study were to determine the prevalence of malnutrition, clinically important weight loss, low muscle strength and nutrition impact symptoms. Secondary aims were to determine factors associated with malnutrition and unintentional weight loss, and to investigate associations between nutritional status and surgical outcomes (complications and length of stay (LOS)). Further outcomes of the NOURISH point prevalence study, including site-specific nutritional practices, perioperative nutritional intervention and nutritional adequacy post-surgery, will be reported in subsequent publications.

## 2. Materials and Methods

### 2.1. Study Design and Setting

A prospective, observational point prevalence study was conducted at 27 tertiary hospitals across six states in Australia. Ethics approval was obtained from The Peter MacCallum Cancer Centre Ethics Committee (Melbourne VIC, Australia) prior to commencement (LNR/51107/PMCC-2019). Further details of study design, participating sites and methods are reported in the study protocol [17]. The target sample size was set at a minimum of 200 participants to obtain a minimum precision of ±7% for the 95% confidence interval (CI) of malnutrition prevalence [17].

### 2.2. Participants

Study dietitians identified and approached eligible patients prior to, or during their surgical admission. Participants were eligible if they were ≥18 years, a hospital inpatient having received curative intent surgery for UGI cancer including gastrectomy (total, subtotal, distal, partial), pancreatectomy (total, distal, partial, pancreatico-duodenectomy), oesophagectomy or gastro-oesophagectomy), able to consent to participation by English language communication or with the presence of an interpreter, and if they received assessment of nutritional status with subjective global assessment (SGA) by a dietitian within seven days of surgery. Participants were excluded if they had received palliative surgery, or non-oncological UGI surgery, unable to participate in SGA, unable to provide consent including if they were on intravenous opioids at time of consent, or they were unaware of diagnosis of malignancy. Recruitment commenced on 2 September 2019 and ended on 30 May 2020. Data collection was completed on 30 June 2020. All participants provided verbal consent to participate according to the approved ethics statement [17] and received the standard dietetics care of the participating health service throughout this study.

### 2.3. Data Collection

Site investigator dietitians conducted a nutritional assessment within seven days of surgery. Patients were also asked to recall information regarding preoperative weight loss, symptoms and prior dietetics or nutritional intervention. The remainder of data were extracted from the medical records. Baseline characteristics included age, sex, primary language spoken, usual social situation, and postcode of residence to determine metropolitan or regional/rural locality. Clinical data included date of diagnosis, tumour type, site and pathological tumour staging (T stage) [18], details of neoadjuvant chemo/radiotherapy if relevant, surgical procedure and surgical technique.

### 2.4. Outcome Measures

The primary outcome of nutritional status was measured according to the validated SGA, which includes a comprehensive assessment of recent weight loss, symptoms, dietary intake, metabolic requirements, physical and functional status [19]. Current weight was measured using calibrated scales, or was patient reported. Weight on discharge was measured where possible. Patients were asked to recall their weight history from 2 weeks, 1, 3, 6 and >6 months prior to surgery, and if they had lost weight unintentionally prior to surgery. Percentage weight loss was calculated, with clinically important weight loss defined as ≥5% in 6 months [20]. Body mass index (BMI) (in kg/m^2^) was calculated from weight and height and categorised as underweight (<18.5 if age < 65 or <22 if age ≥ 65), healthy weight (18.5–24.9 if age < 65 or 22–27 if age ≥ 65) and overweight/obese (≥25 if age < 65 or >27 if age ≥ 65). Upper body muscle strength was measured using hand grip strength (HGS) dynamometry as per the methodology of the American Society of Hand Therapists (ASHT) () [21]. Thresholds for diagnosis of low muscle strength were <20 kg for women and <30 kg for men, or <18 kg for women and <26 kg for men for participants of Asian background [20]. Participants were asked to report prevalence of gastrointestinal symptoms persisting > 2 weeks prior to surgery that have been impacting their ability to eat (poor appetite, nausea, vomiting, diarrhoea, constipation, pain when eating, taste changes, dry mouth, problems swallowing or early satiety), as well as any reduction in food intake, degree and length of time of reduction. Discharge destination and LOS (days) were recorded from the medical records. Surgical complications were recorded as documented by the medical team. These included sepsis, anastomotic leak, pancreatic fistula, pneumonia/respiratory tract infection, pneumothorax, pressure injury, wound infection, return to theatre, abdominal collection, ileus, chyle leak, gastroparesis and pleural effusion. Total number of complications were recorded and transformed into a binary variable of ‘no complication’ or ‘≥1 complication’ for analysis.

### 2.5. Statistical Analysis

Statistical analyses were conducted using Stata/IC 16.0 software (StataCorp, College Station, TX, USA, 2019). Descriptive statistics included measurements such as frequencies, percentages, mean and standard deviation (SD), and median and interquartile range (IQR). Differences in nutritional outcomes between surgical procedures were determined using Fisher’s exact test. Univariate logistic regression analysis was used to determine demographic and clinical factors associated with malnutrition and clinically significant weight loss prior to surgery. Multivariate logistic regression models were developed to determine factors independently associated with malnutrition and unintentional weight loss. Best models were selected using statistical significance threshold of *p* < 0.05 and goodness of fit $R^{2}$. Multivariate models adjusting for age, surgical procedure, tumour location and tumour stage were developed to determine associations between malnutrition and unintentional weight loss with LOS (continuous outcome, linear regression), and surgical complications (binary outcome, logistic regression).

## 3. Results

Of the 240 patients screened, 217 met the eligibility criteria and of those, 200 consented to participate. Figure 1 reports participant flow.

### 3.1. Baseline and Clinical Characteristics

Baseline and clinical characteristics of the overall cohort, as well as for each surgery type (oesophagectomy, gastrectomy and pancreatectomy procedures) are presented in Table 1. There were no significant differences for surgery type in age or sociodemographic factors. A higher proportion of males underwent gastrectomy than oesophagectomy and pancreatectomy procedures (*p* = 0.002), while a higher proportion of patients undergoing oesophagectomy and gastrectomy received neoadjuvant therapy than those who underwent pancreatectomy (*p* < 0.001).

### 3.2. Nutritional Status, Weight Loss and Muscle Strength

The overall prevalence of malnutrition according to SGA was 42% (*n* = 84), 95% CI (35%,49%). The prevalence of any degree of unintentional weight loss was 65% (*n* = 129), 95% CI (57%,71%), whilst 47% (*n* = 48), 95% CI (37%,57%) of those who completed the HGS had low muscle strength (Table 2). The HGS test was unable to be completed in 49% (*n* = 98) of the cohort due to unavailability of equipment or participants being too unwell to complete this assessment within seven days of surgery. There were no differences in nutritional status, unintentional weight loss or muscle strength between surgery types (Table 2).

### 3.3. Prevalence of Clinically Significant Weight Loss

Median (IQR) percentage weight loss in 1, 3, 6 and >6 months was 1.2% (0%, 3.2%), 4.8% (1.8%, 8.2%), 7.4% (3.8%, 11.6%) and 8.3% (4.3%, 12.5%), respectively. There were no differences in median weight loss between the surgery types for any timeframe (Appendix A), except for 1 month before surgery where patients undergoing pancreatectomy procedures lost more weight than those undergoing oesophagectomy and gastrectomy procedures (2.2% (0.0%, 4.5%) vs. 0.5% (0.0%,2.5%) and 0.0% (0.0%, 1.9%) *p* = 0.021). Figure 2 demonstrates the proportion of patients with clinically significant weight loss of ≥5% in the 3 months and 6 months before surgery. Twenty-seven percent of participants reported ≥10% weight loss in >6 months. There were no differences in prevalence of clinically significant weight loss at any timepoint before surgery between the surgery groups (Appendix A).

### 3.4. Prevalence of Preoperative Nutrition Impact Symptoms

The overall prevalence of reporting at least one preoperative gastrointestinal nutrition impact symptom (Table 2) was 55%, with poor appetite and early satiety being the most reported (37% and 23%, respectively). Patients undergoing pancreatic surgery were more likely to report preoperative diarrhoea (18% vs. 0% oesophagectomy and 8% gastrectomy, *p* = 0.001), whilst patients undergoing oesophagectomy and gastrectomy were more likely to report taste changes (23% and 18% vs. 5%, *p* = 0.005). Problems swallowing were reported in a significantly higher proportion of oesophagectomy patients (20% versus 0% gastrectomy and 2% pancreatectomy, *p* < 0.001), whilst early satiety was reported in 34% of gastrectomy and 23% of pancreatectomy, compared to 14% oesophagectomy procedures (*p* = 0.034). Reduced dietary intake was reported in 49% of the cohort, with the majority (35%) having reduced intake for ≥1 month. Patients undergoing oesophagectomy were more likely to report reduced intake for ≥1 month (44% vs. 31% gastrectomy and 28% pancreatectomy, *p* = 0.021).

### 3.5. Factors Associated with Preoperative Malnutrition and Clinically Significant Weight Loss

On univariate model analysis, factors associated with malnutrition included age, T4 stage, unintentional weight loss of any amount, unintentional weight loss of ≥5% and ≥10% in six months, reduced dietary intake of all degrees and timeframes, and the presence of any GI symptoms (Table 3 and Table 4). On multivariate analysis, age, unintentional weight loss of ≥5% in 6 months, vomiting, reduced solid food intake lasting 2–4 weeks and >1 month were independently associated with preoperative malnutrition (Table 5). Factors associated with clinically significant weight loss on univariate analysis included poor appetite, nausea, vomiting, pain when eating, problems swallowing, early satiety, reduced dietary intake, and degree of solid food intake reduction of <75%, 50%, 25% of usual intake for all timeframes (Table 3). Factors independently associated with unintentional weight loss were poor appetite, and degree of solid food reductions of <75% and <50% of usual intake (Table 5).

### 3.6. Associations between Preoperative Nutritional Status and Clinically Significant Weight Loss, with Surgical Length of Stay and Complications

Using bivariate analysis, patients who were malnourished had an increased surgical LOS of 4 days compared with well-nourished participants (14 days, IQR 8,18 versus 10 days, IQR 8,14, *p* = 0.046). Patients with unintentional weight loss of ≥5% in 6 months also had an increased LOS of 4 days compared with those with ≤ 5% (14 days, IQR 8, 19.3 versus 10 days IQR 8,14, *p* = 0.007). On multivariate model analysis (adjusting for age, surgical procedure, tumour location and stage), malnutrition (Coefficient 3.6, 95% CI 0.1, 7.2, *p* = 0.048) was independently associated with increased LOS, as was unintentional weight loss (Coefficient 4.1, 95% CI 0.5,7.6, *p* = 0.026). There were no differences in prevalence of surgical complications between malnourished or well-nourished participants, or patients with clinically significant weight loss on bivariate, univariate, or multivariate model analysis.

## 4. Discussion

NOURISH is the largest study to assess nutritional status in UGI cancer patients at the time of curative intent surgery using a validated assessment method. The majority of other studies of nutritional status in UGI cancer have been undertaken in ambulatory populations or have been included as part of mixed cancer or gastrointestinal surgical populations [14,22,23,24,25,26,27]. This study is unique as patients were recruited from 27 tertiary health services at the time of curative intent surgery, only major UGI surgical oncology procedures were included, and the validated SGA tool was used to diagnose malnutrition.

### 4.1. Nutritional Status, Weight Loss and Muscle Strength

Overall, 42% of participants were malnourished which is relatively consistent with previous studies investigating nutritional status in UGI cancer cohorts, despite some differences in methodology, population or setting [14,23,24]. Interestingly, there were no differences between malnutrition prevalence rates between surgical procedures or tumour types, indicating that all patients presenting for major UGI cancer surgeries are equally at high risk. In this study, almost one-third (30.4%) of patients had clinically significant weight loss within the three months prior to surgery. This is particularly relevant for patients with oesophageal and gastric tumours undergoing neoadjuvant therapy (which typically spans three months including the post-treatment washout period), as weight loss during chemo/radiotherapy is associated with treatment intolerance and decreased survival [24]. It appears that nutritional interventions provided to participants of this study may not be adequate given the high rates of ongoing weight loss and malnutrition. A lower proportion of patients undergoing pancreatic procedures received preoperative chemotherapy treatment, which is consistent with current clinical practice [28]. However, this group presented with the highest proportion of preoperative weight loss for the 2 week, 1 month and 3 month periods prior to surgery, indicating rapid weight loss that is more likely to be associated with tumour progression and associated symptoms, rather than the effects of chemotherapy. This indicates the need for early dietetics intervention in the hospital outpatient clinic setting for patients undergoing pancreatic surgery, which is typically less resourced than the chemoradiotherapy setting [9].

Although 47% of participants had low muscle strength according to HGS, only 51% of participants were able to complete the test within the seven-day post-operative period, limiting interpretation of these results. Whilst other studies have demonstrated that using HGS for GI surgical patients is a valid method of muscle strength assessment [22,29], these studies have undertaken the test within two-five days of surgery in cohorts consisting largely of general or colorectal GI surgical procedures. This was not often possible in this study due to patients being too unwell post-major UGI surgery. Future studies should attempt to complete the HGS on admission for surgery in UGI patients. However, this was unable to be completed in our study due to funding and staff limitations.

### 4.2. Prevalence of Preoperative Nutrition Impact Symptoms

Over half of the NOURISH cohort experienced nutrition impact symptoms persisting for two or more weeks preoperatively, with symptoms consistent with clinical expectations for each cancer type. This highlights the importance of targeted screening for GI symptoms known to be related to each cancer type and ensuring appropriate intervention. For example, pancreatic enzyme replacement therapy is recommended to improve symptoms of early satiety and diarrhoea in patients with pancreatic cancer [30]. In oesophageal cancer, previous studies have also demonstrated that patients commonly report dysphagia and poor appetite prior to the commencement of preoperative chemo/radiotherapy [31,32]. However, these symptoms are typically thought to improve after neoadjuvant therapy and its associated side effects subside [33]. In this study, dysphagia and pain when eating were still present in 31% of patients with oesophageal cancer at the time of surgery, indicating that symptoms continue for many patients after the cessation of neoadjuvant treatment. This has implications on the provision of nutrition support, as patients may not have as much contact with the dietitian or treating team in between neoadjuvant treatment completion and surgery.

### 4.3. Factors Associated with Malnutrition and Unintentional Weight Loss

Whilst a number of factors were associated with malnutrition on univariate analysis, age, unintentional weight loss of ≥5% in 6 months, vomiting and reduced dietary intake remained independently associated on multivariate analysis. Unintentional weight loss of ≥5% in 6 months is well recognised as a key component of a malnutrition diagnosis across all clinical populations and is a phenotypical criterion of the recently developed Global Leadership Initiative on Malnutrition (GLIM) criteria [20]. Our study confirms that it is associated with higher risk of malnutrition in people with UGI cancer. Previous studies in both surgical and oncological cohorts have identified age as an independent factor for malnutrition [14,34], which can impact on clinical outcomes. A recent study demonstrated that malnutrition in older adults with gastrointestinal cancer was associated with impairments in geriatric assessment measures [35]. This study reinforces recommendations that age should be considered when performing surgical risk stratification, and nutritional intervention and exercise should be provided to improve clinical outcomes [6]. Vomiting and loss of appetite were also independently associated with malnutrition in a study of 4783 cancer patients [34], whilst a higher symptom burden has also been associated with poorer nutritional status in a longitudinal study of oesophago-gastric cancer patients undergoing radical treatment [32]. However, this is the first study, to our knowledge, to investigate symptoms persisting at the time of surgery. Similarly, poor appetite and a reduced food intake were associated with clinically significant weight loss. Current oncological guidelines recommend that gastrointestinal symptoms should be monitored as risk factors for malnutrition [36]. The results of this study further highlight the importance of monitoring symptoms and dietary intake, which may not be captured in some malnutrition screening tools. Given that nutritional intervention should be provided for at least 7–10 days before surgery to have an effect on surgical outcomes [5], screening of symptoms and dietary intake should be conducted as early as possible, and ideally should be repeated at frequent intervals until one week prior to surgery.

### 4.4. Associations between Nutritional Status and Surgical Outcomes

Malnutrition and unintentional weight loss of ≥5% in 6 months were both independently associated with a longer surgical LOS. This finding consolidates previous research demonstrating that malnutrition and unintentional weight loss are key modifiable risk factors for post-operative outcomes and that preoperative nutrition intervention is essential [5]. Malnutrition has been linked to an increased risk of surgical complications previously [23]; however, this was not demonstrated in our study. Although we recorded complications known to be affected by malnutrition, site investigators did not classify complications into severe/non-severe as per the Clavien Dindo grading system [37] due to funding and time constraints. However, analysis of each complication separately did not reveal any significant differences between malnourished and well-nourished participants. Patients who were malnourished may have been more likely to receive perioperative enteral or parenteral nutrition support, which has not been accounted for in this analysis.

### 4.5. Strengths and Limitations

Strengths of this study include the large sample size for a UGI surgical oncology cohort, with representation from public and private hospital settings across six Australian states. The SGA is a validated nutritional assessment tool in both surgical and oncology populations. Limitations include the inability to assess muscle mass using an objective tool, such as computed tomography (CT) analysis, as well as the low number of participants who were able to perform the HGS test post-surgery. Although standardised training was conducted to ensure inter-rater reliability, and all data collectors were experienced clinical dietitians, data collection bias due to multiple assessors cannot be excluded.

## 5. Conclusions

The present study demonstrates that a high proportion of patients undergoing major UGI oncological resections still present with malnutrition or clinically significant weight loss at the time of curative surgery, despite increasing recognition of the importance of preoperative nutritional intervention. Risk factors including age, presence of GI symptoms and decreased food intake prior to surgery should be considered during preoperative nutritional risk stratification in clinical practice and appropriate nutritional intervention should be provided, as outlined by current oncology guidelines. Further research is required to determine if current practice for dietetics intervention prior to UGI surgery has a positive impact on clinical outcomes. This could support prioritization of service delivery improvement initiatives and nutrition research trials in this high-risk group.

## Figures and Tables

**Figure 1 nutrients-13-03349-f001:**
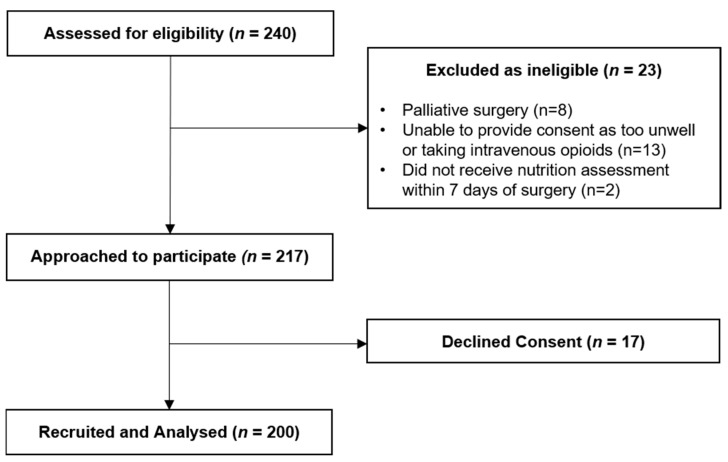
Participant flow diagram.

**Figure 2 nutrients-13-03349-f002:**
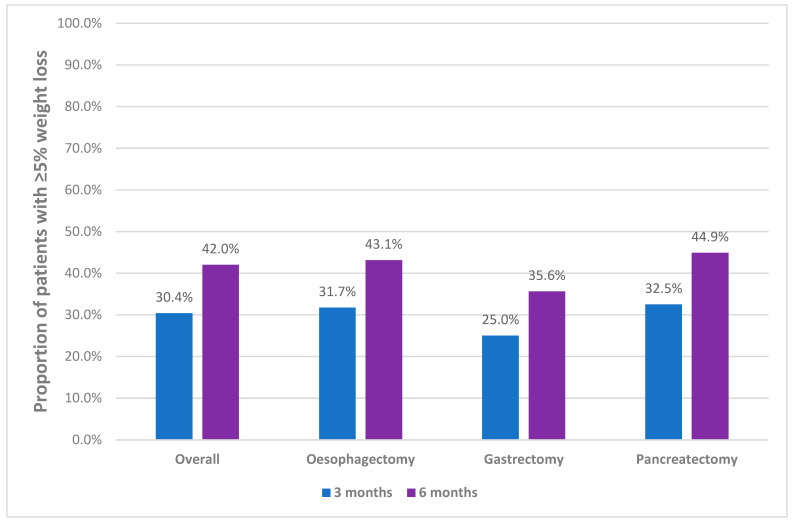
Prevalence of ≥5% weight loss in 3 and 6 months.

**Table 1 nutrients-13-03349-t001:** Demographic, tumour and surgical characteristics by surgery type.

Variables	Overall		Oesophagectomy		Gastrectomy		Pancreatectomy		*p* Value
	*n* = 200		*n* = 66		*n* = 50 ^a^		*n* = 84 ^b^		
Age (Mean, SD)	67	10	66	9	70	11	66	10	0.111
Sex (*n*, %)									**0.002**
Male	117	58.5%	50	75.8%	27	54.0%	40	47.6%	
Female	83	41.5%	16	24.2%	23	46.0%	44	52.4%	
Tumour Location (*n*, %)									NA
Gastric	50	25.0%	1	1.5%	48	96.0%	1	1.2%	
Oesophageal	60	30.0%	60	90.9%	NA	NA	NA	NA	
Pancreatic	55	27.5%	NA	NA	NA	NA	55	65.5%	
Ampullary	17	8.5%	NA	NA	NA	NA	17	20%	
GOJ	7	3.5%	5	7.6%	2	4.0%	NA	NA	
Bile Duct	6	3.0%	NA	NA	NA	NA	6	7.1%	
Duodenal	5	2.5%	NA	NA	NA	NA	5	6.0%	
Tumour Type (*n*, %)									NA
Adenocarcinoma	170	85.0%	54	81.8%	49	98.0%	67	79.8%	
SCC	11	5.5%	11	16.7%	0	0.0%	NA	NA	
GIST	2	1.0%	NA	NA	1	2.0%	1	1.2%	
NET	11	5.5%	NA	NA	0	0.0%	11	13.1%	
Other	6	3.0%	1	1.5%	0	0.0%	5	6.0%	
Intraoperative Tumour Stage (*n*, %)									**<0.001**
T0	15	7.5%	12	18.2%	2	4.0%	1	1.2%	
T1	44	22.0%	20	30.3%	11	22.0%	13	15.5%	
T2	49	24.5%	10	15.2%	8	16.0%	31	36.9%	
T3	63	31.5%	21	31.8%	20	40.0%	22	26.2%	
T4	14	7.0%	0	0.0%	9	18.0%	5	6.0%	
TX	2	1.0%	0	0.0%	0	0.0%	2	2.4%	
Unknown/Unassessed ^c^	13	6.5%	3	4.5%	0	0.0%	10	11.9%	
Received Neoadjuvant Therapy (n, %)									**<0.001**
No	106	53.3%	17	25.8%	15	30.6%	74	88.1%	
Yes	93	46.7%	49	74.2%	34	69.4%	10	11.9%	
Type of Neoadjuvant Therapy (*n*, %) ^d^									**<0.001**
Chemotherapy	52	55.9%	10	20.4%	32	94.1%	10	100.0%	
Chemotherapy and Radiotherapy	41	44.1%	39	79.6%	2	5.9%	0	0.0%	
Completed Neoadjuvant Therapy (*n*, %) ^d^									
No	3	3.2%	0	0.0%	3	8.9%	0	0.0%	
Yes	88	94.6%	49	100.0%	29	85.3%	10	100.0%	
Unknown	2	2.2%	0	0.0%	2	5.8%	0	0.0%	
Location of Residence (*n*, %)									0.482
Metropolitan	143	71.5%	45	68.2%	39	78.0%	59	70.2%	
Rural/Regional	57	28.5%	21	31.8%	11	22.0%	25	29.8%	
Social Situation (*n*, %)									0.872
Lives Alone	45	22.5%	13	19.7%	10	20.0%	22	26.2%	
Lives with Family/Carer	152	76.0%	52	78.8%	39	78.0%	61	72.6%	
Lives in Shared Accommodation	0	0.0%	0	0.0%	0	0.0%	0	0.0%	
Lives in Residential Care	3	1.5%	1	1.5%	1	2.0%	1	1.2%	
Surgical Technique (*n*, %)									**<0.001**
Open	187	93.5%	53	80.3%	50	100.0%	84	100.0%	
Laparoscopic/Minimally Invasive	13	6.5%	13	19.7%	0	0.0%	0	0.0%	

^a^ Includes total, subtotal, partial and distal gastrectomy. ^b^ Includes total, distal, partial, pancreatico-duodenectomy. ^c^ Incomplete information from medical records. ^d^ Presented as a proportion of participants who responded ‘yes’ to neoadjuvant therapy (*n* = 93). SCC = squamous cell carcinoma, GIST = gastrointestinal stromal tumour, NET = neuroendocrine tumour, and GOJ = gastro-oesophageal. Bolded *p* values indicate statistical significance.

**Table 2 nutrients-13-03349-t002:** Nutritional status and nutrition impact on symptoms by surgery type.

Variable	Overall		Oesophagectomy		Gastrectomy		Pancreatectomy		*p* Value
	*n* = 200		*n* = 66		*n* = 50		*n* = 84		
SGA Rating (*n*, %)									0.689
A No Malnutrition	116	58.0%	40	60.6%	28	56.0%	48	57.1%	
B Mild/Moderate Malnutrition	79	39.5%	24	36.4%	22	44.0%	33	39.3%	
C Severe Malnutrition	5	2.5%	2	3.0%	0	0.0%	3	3.6%	
Overall Nutritional Status (*n*, %)									0.864
Well Nourished (SGA A)	116	58.0%	40	60.6%	28	56.0%	48	57.1%	
Malnourished (SGA B/C)	84	42.0%	26	39.4%	22	44.0%	36	42.9%	
Unintentional Weight Loss (*n*, %)									0.645
No	71	35.5%	24	36.4%	20	40.0%	27	32.1%	
Yes	129	64.5%	42	63.6%	30	60.0%	57	67.9%	
Low Muscle Strength (*n*, %)									0.075
No	54	27.0%	23	34.8%	9	1.08%	22	26.2%	
Yes	48	24.0%	12	18.2%	16	32.0%	20	23.8%	
Not Completed	98	49.0%	31	47.0%	25	50.0%	42	50.0%	
BMI (kg/m^2^) (Mean, SD)	27.3	5.6	27.3	5.1	27.5	6.1	27.2	5.7	0.917
BMI Categories (*n*, %)									0.214
Underweight	20	10.0%	3	4.5%	8	16.0%	9	10.7%	
Normal Weight	75	37.5%	30	45.5%	17	34.0%	28	33.3%	
Overweight/Obese	105	52.5%	33	50.0%	25	50.0%	47	56.0%	
Reduced Dietary Intake before Surgical Admission (*n*, %)									0.099
No	101	50.5%	34	51.5%	31	62.0%	36	42.9%	
Yes	99	49.5%	32	48.5%	19	38.0%	48	57.1%	
Degree of Reduction in Solid Food Intake (*n*, %)									0.602
>75% of Usual Intake	27	13.5%	10	15.2%	6	12.0%	11	13.1%	
≤75% of Usual Intake	34	17.0%	11	16.7%	6	12.0%	17	20.2%	
≤50% of Usual Intake	31	15.5%	8	12.1%	6	12.0%	17	20.2%	
≤25% of Usual Intake	7	3.5%	3	4.5%	1	2.0%	3	3.6%	
No Reduction in Intake	101	50.5%	34	51.5%	31	62.0%	36	42.9%	
Length of Time of Reduced Dietary Intake (*n*, %)									**0.021**
<1 Week	2	1.0%	2	1.5%	0	0.0%	1	1.2%	
1–2 Weeks	11	5.5%	2	1.5%	3	6.0%	7	8.3%	
2–4 Weeks	13	6.5%	2	1.5%	1	2.0%	11	13.1%	
≥1 Month	69	34.5%	29	43.9%	14	28.0%	26	31.0%	
No Reduction in Intake	105	52.5%	34	51.5%	32	64.0%	39	46.4%	
Symptoms Persisting >2 Weeks Prior to Surgery Impacting Ability to Eat									
Poor Appetite									0.062
No	126	63.0%	49	74.2%	30	60.0%	47	56.0%	
Yes	74	37.0%	17	25.8%	20	40.0%	37	44.0%	
Nausea									0.307
No	172	86.0%	60	90.9%	43	86.0%	69	82.1%	
Yes	28	14.0%	6	9.1%	7	14.0%	15	17.9%	
Vomiting									0.952
No	187	93.5%	62	93.9%	47	94.0%	78	92.9%	
Yes	13	6.5%	4	6.1%	3	6.0%	6	7.1%	
Diarrhoea									**0.001**
No	181	90.5%	66	100.0%	46	92.0%	69	82.1%	
Yes	19	9.5%	0	0.0%	4	8.0%	15	17.9%	
Constipation									0.382
No	193	96.5%	62	93.9%	49	98.0%	82	97.6%	
Yes	7	3.5%	4	6.1%	1	2.0%	2	2.4%	
Pain When Eating									0.966
No	178	89.0%	59	89.0%	44	88.0%	75	89.3%	
Yes	22	11.0%	7	11.0%	6	12.0%	9	10.7%	
Taste Changes									**0.005**
No	172	86.0%	51	77.3%	41	82.0%	80	95.2%	
Yes	28	14.0%	15	22.7%	9	18.0%	4	4.8%	
Dry Mouth									0.738
No	191	95.5%	64	97.0%	47	94.0%	80	95.2%	
Yes	9	4.5%	2	3.0%	3	6.0%	4	4.8%	
Problems Swallowing									**<0.001**
No	185	92.5%	53	80.3%	50	100.0%	82	97.6%	
Yes	15	7.5%	13	19.7%	0	0.0%	2	2.4%	
Early Satiety									**0.034**
No	155	77.5%	57	86.4%	33	66.0%	65	77.4%	
Yes	45	22.5%	9	13.6%	17	34.0%	19	22.6%	
No Problems Reported									0.972
No	110	55.0%	36	54.5%	27	54.0%	47	56.0%	
Yes	90	45.0%	30	45.5%	23	46.0%	37	44.0%	

SGA = subjective global assessment, SD = standard deviation, and BMI = body mass index. Bolded *p* values indicate statistical significance.

**Table 3 nutrients-13-03349-t003:** Demographic and clinical factors associated with malnutrition and clinically significant weight loss (≥5% in 6 months) prior to surgery (*n* = 200).

	Malnutrition					Unintentional Weight Loss ≥ 5% in 6 Months				
	Bivariate Analysis (Fisher’s Exact)			Univariate Logistic Model		Bivariate Analysis (Fisher’s Exact)			Univariate Logistic Model	
Variable	WN *n* (%)	MN *n* (%)	*p* Value	OR (95% CI)	*p* Value	No *n* (%)	Yes *n* (%)	*p* Value	OR (95% CI)	*p* Value
Age			**0.012**					0.549		
<65	54 (69.2)	24 (30.8)		1.0 (ref)		45 (60.8)	29 (39.2)		1.0 (ref)	
≥65	62 (50.8)	60 (49.2)		2.2 (1.2, 4.0)	**0.011**	64 (56.1)	50 (43.9)		1.2 (0.7, 2.2)	0.526
Sex (*n*, %)			0.082					0.654		
Male	74 (63.2)	43 (36.8)		1.0 (ref)		66 (59.5)	45 (40.5)		1.0 (ref)	
Female	42 (50.6)	41 (49.4)		1.7 (0.9, 3.0)	0.075	43 (55.8)	34 (44.2)		1.2 (0.6, 2.1)	0.621
Surgery Type			0.864					0.588		
Oesophagectomy	40 (60.6)	26 (39.4)		1.0 (ref)		37 (56.9)	28 (43.1)		1.0 (ref)	
Gastrectomy	28 (56.0)	22 (44.0)		1.2 (0.6, 2.5)	0.618	29 (64.4)	16 (35.6)		0.7 (0.3, 1.6)	0.429
Pancreatectomy	48 (57.1)	36 (42.9)		1.2 (0.6, 2.2)	0.669	43 (55.1)	35 (44.9)		1.1 (0.6, 2.1)	0.830
Tumour Location			0.926					0.380		
Bile Duct	4 (66.7)	2 (33.3)		1.0 (ref)		4 (66.7)	2 (33.3)		1.0 (ref)	
Gastric	28 (56.0)	22 (44.0)		1.6 (0.3, 9.4)	0.620	29 (63.0)	17 (37.0)		1.2 (0.2, 7.1)	0.862
Oesophageal	37 (61.7)	23 (38.3)		1.2 (0.2, 7.3)	0.810	34 (56.7)	26 (43.3)		1.5 (0.3, 9.0)	0.638
Pancreatic	32 (58.2)	23 (41.8)		1.4 (0.2, 8.5)	0.689	29 (56.9)	22 (43.1)		1.5 (0.3, 9.1)	0.657
Ampullary	10 (58.8)	7 (41.2)		1.4 (0.2, 9.9)	0.736	10 (62.5)	6 (37.5)		1.2 (0.2, 8.7)	0.857
Duodenal	2 (40.0)	3 (60.0)		3 (0.3, 35.3)	0.383	0 (0)	4 (100)		Empty	
GOJ	3 (42.9)	4 (57.1)		2.7 (0.3, 25.6)	0.396	3 (60)	2 (40)		1.3 (0.1, 15.7)	0.819
Tumour Type			0.335					0.912		
Adenocarcinoma	95 (55.9)	75 (44.1)		1.6 (0.3, 8.9)	0.604	91 (57.2)	68 (42.8)		0.8 (0.2, 3.8)	0.726
SCC	6 (54.5)	5 (45.5)		1.7 (0.2, 13.2)	0.629	7 (63.6)	4 (36.4)		0.6 (0.1, 4.3)	0.587
GIST	2 (100)	0 (0)		Empty		1 (50)	1 (50)		1.0 (0.1, 24.6)	1.00
NET	9 (81.8)	2 (18.2)		0.4 (0.0, 4.4)	0.487	7 (70)	3 (30)		0.4 (0.1, 3.5)	0.428
Other	4 (66.7)	2 (33.3)		1.0 (ref)		3 (50)	3 (50)		1.0 (ref)	
Tumour Stage			**0.022**					0.509		
T0	10 (66.7)	5 (33.3)		1.0 (ref)		9 (64.3)	5 (35.7)		1.0 (ref)	
T1	31 (70.5)	13 (29.5)		0.8 (0.2, 2.9)	0.783	28 (65.1)	15 (34.9)		1.0 (0.3, 3.4)	0.955
T2	29 (59.2)	20 (40.8)		1.4 (0.4, 4.7)	0.604	25 (55.6)	20 (44.4)		1.4 (0.4, 5.0)	0.565
T3	29 (46.0)	34 (54.0)		2.3 (0.7, 7.6)	0.158	31 (51.7)	29 (48.3)		1.7 (0.5, 5.6)	0.397
T4	4 (28.6)	10 (71.4)		5.0 (1.0, 24.3)	**0.046**	5 (41.7)	7 (58.3)		2.5 (0.5, 12.3)	0.253
Neoadjuvant Therapy			0.774					0.882		
No	60 (56.6)	46 (43.4)		1.0 (ref)		57 (58.8)	40 (41.2)		1.0 (ref)	
Yes	55 (59.1)	38 (40.9)		0.9 (0.5, 1.6)	0.718	51 (56.7)	39 (43.3)		1.1 (0.6, 2.0)	0.772
Type of Neoadjuvant			0.205					0.089		
Chemotherapy	34 (65.4)	18 (34.6)		1.0 (ref)		32 (65.3)	17 (34.7)		1.0 (ref)	
Chemotherapy and Radiotherapy	21 (51.2)	20 (48.8)		0.6 (−0.3, 1.4)	0.169	19 (46.3)	22 (53.7)		2.2 (0.9, 5.1)	0.072
Completed Neoadjuvant			1.000					1.000		
No	2 (66.7)	1 (33.3)		1.0 (ref)		2 (66.7)	1 (33.3)		1.0 (ref)	
Yes	51 (58.0)	37 (42.0)		1.5 (0.1, 16.6)	0.765	47 (55.3)	38 (44.7)		1.6 (0.1, 18.5)	0.699
Location of Residence			0.428					0.137		
Rural/Regional	36 (63.2)	21 (36.8)		1.0 (ref)		35 (67.3)	17 (32.7)		1.0 (ref)	
Metropolitan	80 (55.9)	63 (44.1)		1.3 (0.7, 2.5)	0.352	74 (54.4)	62 (45.6)		1.7 (0.9, 3.4)	0.111
Social Situation			0.132					0.863		
Lives with Family or Carer	94 (61.8)	58 (38.2)		1.0 (ref)		85 (59)	59 (41)		1.0 (ref)	
Lives Alone	21 (46.7)	24 (53.3)		1.8 (0.9, 3.6)	0.07	23 (54.8)	19 (45.2)		1.2 (0.6, 2.4)	0.622
Lives in Residential Care	1 (33.3)	2 (66.7)		3.2 (0.3, 36.5)	0.341	1 (50)	1 (50)		1.4 (0.1, 23.5)	0.798

WN = well nourished, MN = malnourished, OR = odds ratio, CI = confidence interval, SCC = squamous cell carcinoma, GIST = gastrointestinal stromal tumour, NET = neuroendocrine tumour, and GOJ = gastro-oesophgeal. Bolded *p* values indicate statistical significance. Ref = reference value used in the model.

**Table 4 nutrients-13-03349-t004:** Nutrition-related factors associated with malnutrition and clinically significant weight loss (≥5% in 6 months) prior to surgery (*n* = 200).

	Malnutrition					UnintentionalNINTENTIONAL Weight Loss ≥5% in 6 Months				
	Bivariate Analysis (Fisher’s Exact)			Univariate Logistic Model		Bivariate Analysis (Fisher’s Exact)			Univariate Logistic Model	
Variable	No *n* (%)	Yes *n* (%)	*p* Value	OR (95% CI)	*p* Value	No *n* (%)	Yes *n* (%)	*p* Value	OR (95% CI)	*p* Value
BMI			**<0.001**					**<0.001**		
Normal Weight	33 (44.0)	42 (56.0)		1.0 (ref)		37 (53.6)	32 (46.4)		1.0 (ref)	
Underweight	3 (85.0)	17 (85.0)		4.5 (1.2, 16.5)	**0.025**	3(16.7)	15 (83.3)		5.8 (1.5, 21.8)	**0.010**
Overweight/Obese	80 (76.2)	25 (23.8)		0.2 (0.1, 0.5)	**<0.001**	69 (68.3)	21 (31.7)		0.5 (0.3, 1.1)	0.053
GI Symptoms										
Unchecked				1.0 (ref)					1.0 (ref)	
Poor Appetite	17 (23.0)	57 (77.0)	**<0.001**	12.3 (6.2, 24.5)	**<0.001**	18 (26.9)	49 (73.1)	**<0.001**	8.3 (4.2, 16.3)	**<0.001**
Nausea	8 (28.6)	20 (71.4)	**0.001**	4.2 (1.8, 10.1)	**0.001**	7 (26.9)	19 (73.1)	**0.001**	4.6 (1.8, 11.6)	**0.001**
Vomiting	1 (7.7)	12 (92.3)	**<0.001**	19.2 (2.4, 150.6)	**0.005**	3(23.1)	10 (76.9)	**0.016**	5.1 (1.4, 19.3)	**0.016**
Diarrhoea	4 (21.1)	15 (78.9)	**0.001**	6.1 (1.9, 19.1)	**0.002**	6 (35.3)	11 (54.7)	0.069	2.8 (1.0, 7.9)	0.054
Constipation	1 (14.3)	6 (85.7)	**0.043**	8.8 (1.0, 74.9)	**0.046**	2 (28.6)	5 (71.4)	0.133	3.6 (0.7, 19.1)	0.131
Pain When Eating	8 (36.4)	14 (63.6)	**0.039**	2.7 (1.1, 6.8)	**0.034**	4 (21.1)	15 (78.9)	**0.001**	6.2 (2.0, 19.4)	**0.002**
Taste Changes	10 (35.7)	18 (63.3)	**0.013**	2.9 (1.3, 6.6)	**0.012**	13 (48.1)	14 (51.9)	0.296	1.6 (0.7, 3.6)	0.266
Dry Mouth	2 (22.2)	7 (77.8)	**0.037**	5.2 (1.0, 25.6)	**0.044**	4 (50)	4 (50)	0.722	1.4 (0.3, 5.8)	0.642
Problems Swallowing	3 (20.0)	12 (80.0)	**0.002**	6.3 (1.7, 23.0)	**0.006**	4 (28.6)	10 (71.4)	**0.025**	3.8 (1.2, 12.6)	**0.029**
Early Satiety	16 (35.6)	29 (64.4)	**0.001**	3.3 (1.6, 6.6)	**0.001**	17 (42.5)	23 (57.5)	**0.031**	2.2 (1.1, 4.5)	**0.027**
Any LOW			**<0.001**			NA			NA	
No	67 (94.4)	4 (5.6)		1.0 (ref)	**<0.001**					
Yes	49 (38.0)	80 (62.0)		27.3 (9.4, 79.7)						
LOW ≥5% in 6 Months			**<0.001**		**<0.001**	NA			NA	
No	29 (74.4)	10 (25.6)		1.0 (ref)						
Yes	14 (21.5)	51 (78.5)		29.1 (13.1, 64.6)						
LOW ≥ 10% in 6 Months			**<0.001**		**<0.001**	NA			NA	
No	40 (58.0)	29 (42.0)		1.0 (ref)						
Yes	3 (8.6)	32 (91.4)		40.0 (11.6, 138.1)						
Reduced Dietary Intake			**<0.001**					**<0.001**		
No	84 (83.2)	17 (16.8)		1.0 (ref)		75 (77.3)	22 (22.7)		1.0 (ref)	
Yes	32 (32.3)	67 (67.7)		10.3 (5.3, 20.2)	**<0.001**	34 (37.4)	57 (62.6)		5.7 (3.0, 10.8)	**<0.001**
Degree of Reduction in Solid Food Intake			**<0.001**					**<0.001**		
No Reduction in Intake	84 (83.2)	17 (16.8)		1.0 (ref)		75 (77.3)	22 (22.7)		1.0 (ref)	
>75% of Usual Intake	17 (63.0)	10 (37.0)		2.9 (1.1, 7.4)	**0.026**	16 (69.6)	7 (30.4)		1.5 (0.6, 4.1)	0.437
≤75% of Usual Intake	8 (23.5)	26 (76.5)		16.1 (6.2, 41.5)	**<0.001**	10 (32.3)	21 (67.7)		7.2 (2.9, 17.4)	**<0.001**
≤50% of Usual Intake	5 (16.1)	26 (83.9)		25.7 (8.6, 76.4)	**<0.001**	6 (20)	24 (80)		13.6 (5.0, 37.6)	**<0.001**
≤25% of Usual Intake	2 (28.6)	5 (71.4)		12.4 (2.2, 69.1)	**0.004**	2 (28.6)	5 (71.4)		8.5 (1.6, 47.0)	**0.014**
Length of Time of Reduced Dietary Intake			**<0.001**					**<0.001**		
No Reduction in Intake	88 (83.8)	17 (16.2)		1.0 (ref)		78 (77.2)	23 (22.8)		1.0 (ref)	
<1 Week	2 (100.0)	0 (0)		Empty		1 (100)	0 (0)		(empty)	
1–2 Weeks	5 (45.5)	6 (54.5)		6.2 (1.7, 22.7)	**0.006**	4 (36.4)	7 (63.6)		5.9 (1.6, 22.1)	**0.008**
2–4 Weeks	5 (38.5)	8 (61.5)		8.3 (2.4, 28.4)	**0.001**	6 (46.2)	7 (53.8)		4.0 (1.2, 13.0)	**0.023**
≥1 Month	16 (23.2)	53 (76.8)		17.1 (8.0, 36.8)	**<0.001**	20 (32.3)	42 (67.7)		7.1 (3.5, 14.4)	**<0.001**

OR = odds ratio, CI = confidence interval, GI = Gastrointestinal and LOW = loss of weight. Bolded *p* values indicate statistical significance. Ref = reference value used in the model.

**Table 5 nutrients-13-03349-t005:** Factors independently associated with malnutrition and weight loss ≥ 5% by multivariate analysis (*n* = 200).

Variable	Malnutrition OR (95% CI)	*p* Value	Unintentional Weight Loss ≥ 5%OR (95% CI)	*p* Value
Age ≥ 65	4.1 (1.5, 11.5)	**0.008**		
LOW ≥ 5% in 6 Months	28.7 (10.5, 78.6)	**<0.001**		
Length of Time of Reduced Intake				
2–4 Weeks	7.4 (1.3, 43.5)	**0.026**		
≥1 Month	7.7 (2.7, 22.0)	**<0.001**		
Degree of reduction in solid food intake				
≤75% of Usual Intake			3.3 (1.2, 9.2)	**0.02**
≤50% of Usual Intake			4.9 (1.5, 15.6)	**0.008**
Nutrition Impact Symptoms				
Vomiting	17.1 (1.4, 207.6)	**0.025**		
Poor Appetite			3.7 (1.6, 8.4)	**0.002**

OR = odds ratio, CI = confidence interval, and LOW = loss of weight. Bolded *p* values indicate statistical significance.

## Data Availability

Not applicable.

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
