# Peer review of "Assessment of Nutritional Status and Nutrition Impact Symptoms in Patients Undergoing Resection for Upper Gastrointestinal Cancer: Results from the Multi-Centre NOURISH Point Prevalence Study"

_nutrients, 2021, doi:10.3390/nu13103349_

Round 1
Reviewer 1 Report
“Assessment of nutritional status and nutrition impact symptoms in patients undergoing resection for upper gastrointestinal cancer: results from the multi-centre NOURISH point prevalence study” by Irene Deftereosa et al. This is an interesting/important work for the readers of Nutrients and beyond.
The Authors had carried out the study to find out the prevalence of malnutrition, weight loss, and other nutrition-related symptoms. Besides the principal aims, the Authors also identified the key factors associated with malnutrition and weight loss. The Authors had used appropriate study design, data collection/harmonization methods for building the statistical models. This is a well-written article and deserves publication in Nutrients.
I have a couple of minor comments:
- Table 1: “Received Neoadjuvant Therapy (n, %)” values don’t add up to 200? (No: 106, Yes: 93).
- Can the Authors check that the % listed in Table2 adds up to 100%? For example, look at the “Degree of reduction in solid food intake (n, %)” section.
- Table 4 has formatting issues. I am assuming editors can fix this in the final version?
Author Response
Many thanks for your review and for reviewing the details on the table to find these errors. We have addressed your comments below:
- Table 1: we have checked and edited the values, and also changed the percentages to exact percentages. Please note there are no tracked changes as the entire values of the table have been changed to exact percentages.
- Table 2: we have checked and edited the values, and also changed the percentages to exact percentages. Please note there are no tracked changes as the entire values of the table have been changed to exact percentages.
- Table 4: We agree, it would be better if this table (and table 3) was presented in landscape format. We will leave this to the editing team to format in the version for proof reading.
Reviewer 2 Report
Overall good job and very relevant.
Author Response
Thank you for your review.
Reviewer 3 Report
Dear authors, this is a very well-written manuscript which stresses the importance of nutritional support both pre and postoperative. And how important basic factors are, not only the high-technique surgery methods. Congratulations. I have nothing to critisize about this paper.
Author Response
Thank you for your review.